# A Dual Regulatory Mechanism of Hormone Signaling and Fungal Community Structure Underpin Dendrobine Accumulation in *Dendrobium nobile*

**DOI:** 10.3390/biom15101366

**Published:** 2025-09-26

**Authors:** Yongxia Zhao, Nian Xiong, Xiaolong Ji, Dongliang Zhang, Qi Jia, Lin Qin, Xingdong Wu, Daopeng Tan, Jian Xie, Yuqi He

**Affiliations:** Guizhou Engineering Research Center of Industrial Key-Technology for Dendrobium Nobile, Guizhou Engineering Research Center for Orchid Medicinal Plant Breeding and Efficient Application, Zunyi Medical University, Zunyi 563002, China; zhaoyongxia@zmu.edu.cn (Y.Z.); jiaqi@zmu.edu.cn (Q.J.); qinlin@zmu.edu.cn (L.Q.); wuxingdong@zmu.edu.cn (X.W.); tandp@zmu.edu.cn (D.T.)

**Keywords:** *Dendrobium nobile*, hormone pathways, endophytic fungal communities, FUNGuild

## Abstract

(1) Objective: The biosynthesis of medicinal secondary metabolites in *Dendrobium nobile* Lindl. is regulated by complex environmental, hormonal, and microbial interactions. However, the mechanisms by which subtle variations in plant elevation shape metabolite accumulation through plant–microbe–hormone networks remain largely unexplored. (2) Methods: We conducted a multi-omics investigation of *D. nobile* cultivated under simulated wild conditions at four elevation gradients (347–730 m) in Chishui, China. High-throughput transcriptome sequencing and ITS-based fungal community profiling were combined with hormone quantification and functional prediction (FUNGuild), enabling integrated analysis of hormone pathway activation, microbial structure–function dynamics, and dendrobine levels. (3) Reults: This study systematically investigated *D. nobile* cultivated under simulated wild conditions across four elevation gradients (347–730 m) in the Danxia region of Chishui, China. We identified a dual regulatory mechanism underlying the elevation-dependent accumulation of dendrobine alkaloids, involving both plant hormone signaling and endophytic fungal communities. Transcriptomic analyses revealed coordinated upregulation of key hormone pathway genes, including DELLA, PYR/PYL, SnRK2, COI1-JAZ-MYC2, and NPR1-TGA, particularly in CY01Y samples at 670 m elevation from ChiYan base in Chishui city, which corresponded to the highest dendrobine content. Concurrently, functional prediction of the ITS-based fungal sequencing data revealed that CY01Y harbored a stable, functionally enriched fungal community dominated by saprotrophs, fungal parasites, and plant pathogens. (4) Conclusions: Through integrative hormone profiling, gene expression, and microbial function analysis, we propose that elevation-induced environmental cues reshape hormone pathways both directly and indirectly via microbial feedback. Specific microbial taxa were identified as potential modulators of hormone signaling and secondary metabolism. The coordinated interaction between plant hormones and endophytic fungi supports a hormone–microbiome–metabolite network that dynamically regulates dendrobine biosynthesis in response to micro-elevation variation.

## 1. Introduction

*Dendrobium nobile* Lindl. is a pharmacopeial medicinal plant with profound historical importance in traditional medicine, documented as early as Shennong Bencao Jing and Bencao Jing Ji Zhu [1], highlighting its substantial medicinal value. Notably, the exceptional quality with higher yield, better appearance and higher ingredients content of *D. nobile* cultivated in the Chishui region of Guizhou has gained recognition because of the region’s unique geographical and ecological conditions. This distinctive environment plays a key role in maintaining the authenticity of locally grown *D. nobile*, which is widely acknowledged in both traditional medicine and modern scientific research. The authenticity of medicinal materials is a critical criterion for evaluating their “superior quality and high efficacy,” with ecological factors being pivotal in determining the overall quality of the plant material [2]. These ecological factors can be broadly categorized into abiotic and biotic components.

The Danxia landform in the Chishui region is a UNESCO World Heritage Site [3] and is characterized by the widespread presence of Danxia stones along slopes of varying elevations. These slopes form diverse semi-wild cultivation environments for *D. nobile*, which are all located within the Chishui region of Zunyi, Guizhou Province. While these sites share comparable temperature, humidity, and overall climatic conditions, elevation stands out as a critical abiotic variable among them. Our previous studies demonstrated a significant positive correlation between elevation and the accumulation of dendrobine, a key bioactive sesquiterpene alkaloid in *D. nobile* [4]. Similar altitude-dependent accumulation of secondary metabolites has been observed in other medicinal plants. For example, relatively high altitudes increase the synthesis of antioxidant secondary metabolites in *Rhodiola linearifolia* Boriss [5]. Moreover, protochlorophyllide (Pchlide) accumulation has been shown to be positively correlated with elevation in *Arabidopsis*, *Solanum habrochaites*, *Solanum cheesmaniae*, and *Brachypodium distachyon* [6]. Even in coniferous species such as Lodgepole pine, the concentrations of both primary and secondary metabolites, especially terpenoids, increase with increasing altitude [7]. These studies collectively confirm that the accumulation of metabolites in many plants is closely linked to the elevation of the cultivation site, providing strong evidence that the elevation variation in the slopes in Chishui significantly affects the metabolic products of *D. nobile*.

In addition to abiotic conditions, biotic factors, particularly microbial communities, play essential roles in regulating plant growth and secondary metabolism. Recent research highlights the pivotal role of endophytic microbiomes in modulating plant physiological processes and specialized metabolite biosynthesis [8]. The relationship between plants and endophytes resembles that between the human gut microbiome and its host [9]. Importantly, many environmental factors not only directly affect plants but also indirectly regulate plant growth dynamics and metabolic flux by influencing horizontal transmission among microorganisms and reshaping the structure of the endophytic microbiome [10]. This indirect mode of regulation tends to be more subtle and gentle, possibly granting the plant enhanced environmental adaptability [11]. The mechanisms underpinning such adaptations are closely tied to the functional traits and lifestyles of different microbial taxa, which interact with and modulate the plant’s immune and defense systems [8]. This insight ecologically constructs an intrinsic connection between abiotic factors (elevation) and biotic factors (endophytic microbiomes). Moreover, specific metabolites synthesized by the host plant can provide feedback and influence the structure of the recruited endophytic microbial communities [12].

Dendrobine, a typical sesquiterpene alkaloid [13], is a classic chemical defense compound synthesized by *D. nobile*. Therefore, there is an inherent reciprocal relationship between the biosynthesis of dendrobine and the recruited endophytic fungal community. This complex, multidimensional interaction network between “plant–microbe” and “microbe–microbe” that finely regulates the synthesis and accumulation of dendrobine, the core defense metabolite of this medicinal plant, provides an important biological foundation for the authenticity of *D. nobile* in the Chishui region.

Thus, this study focuses on the *D. nobile* from four semi-wild cultivation bases named YJ01Y, CY01Y, KX01Y, and ZS01Y, which are situated at different elevations (730 m, 670 m, 548 m, and 347 m) in the Chishui region. By employing sample collection and RNA-Seq sequencing of *D. nobile* stems from these bases, coupled with endophytic fungal diversity analysis and FUNGuild functional prediction, we aimed to reveal the relationship between the composition and function of the fungal communities in the stems of *D. nobile* and the elevation of the semi-wild cultivation slopes, as well as their impact on the quality of *D. nobile*.

## 2. Materials and Methods

### 2.1. Sampling Collection

Stems of *D. nobile* were collected from four groups corresponding to the semi-wild cultivation base locations YJ01Y, CY01Y, KX01Y, and ZS01Y, which are situated at different elevations (730 m, 670 m, 548 m, and 347 m) in Chishui city, Guizhou. These samples were obtained from wild-simulated cultivation on Danxia rocks at the ChiYan (CY01Y), YunJi (YJ01Y), KaiXuan (KX01Y), and ZhuanShi (ZS01Y) bases via a systematic sampling approach. Semi-wild cultivation refers to the practice of transplanting artificially propagated *D. nobile* seedlings onto Danxia rock formations in natural settings, where they are managed only minimally. This approach allows the species to grow and reproduce in conditions that closely mimic its native habitat. Among these, ChiYan (CY01Y) and YunJi (YJ01Y) can also be referred to as high-elevation bases, while KaiXuan (KX01Y) and ZhuanShi (ZS01Y) are known as low-elevation bases based on their elevation. A total of six to eight biological replicates were included for each group. The collected samples were divided into two sections for separate analysis, one for transcriptome sequencing and the other for endophytic fungal sequencing, as outlined in a previous study [10].

### 2.2. RNA Sequencing

Total RNA was extracted from 24 *D. nobile* stems, which represent the four base locations. According to the kit protocol (RNAprep Pure Plant Total RNA Extraction Kit (DP432) (TIANGENG, Beijing, China), efficiently lyse cells and inactivate RNases using a proprietary buffer. Bind RNA specifically to a silica membrane in a spin column while removing contaminants like proteins and DNA. Wash away impurities and elute pure RNA with RNase-free water. RNA integrity was assessed via the RNA Nano 6000 Assay Kit (Agilent, Santa Clara, CA, USA) on a Bioanalyzer 2100 system. mRNA was isolated from the total RNA pool via poly-T oligo-attached magnetic beads. Following RNA evaluation, cDNA synthesis was carried out via random hexamer primers and M-MuLV Reverse Transcriptase (RNase H-). The second strand of cDNA was subsequently synthesized via DNA polymerase I and RNase H. cDNA fragments in the 370–420 bp range were subsequently purified and selected via the AMPure XP system (Beckman Coulter, Beverly, MA, USA). Finally, the library preparations were sequenced on an Illumina NovaSeq platform.

### 2.3. Sequencing of Endophytic Fungi

Total genomic DNA was extracted from 32 stems samples of *D. nobile* (eight biological replicates per group) representing four different semi-wild cultivation bases via the CTAB/SDS method. The internal transcribed spacer (ITS) region was amplified with specific primers (forward: GGAAGTAAAAGTCGTAACAAGG, reverse: GCTGCGTTCTTCATCGATGC), each associated with a unique barcode. Library construction was performed, and sequencing was carried out on an Illumina NovaSeq platform. The 250 bp paired-end reads were generated after amplification, purification, and library construction, following the protocol: amplification protocol: 98 °C (1 min) → (98 °C (10 s) → 50 °C (30 s) → 72 °C (30 s) × 30 cycles) → 72 °C (5 min). The NEBNext^®^ Ultra™ II DNA Library Prep Kit (Cat No. E7645) (New England Biolabs, Inc., Hitchin, UK) was used for library preparation, while the Qubit^®^ 2.0 Fluorometer (Thermo Scientific, Waltham, MA, USA) and the Agilent Bioanalyzer 2100 system were used for quality assessment of the library. The endophytic fungal communities in *D. nobile* stems from different habitats were subsequently characterized.

### 2.4. RNA-Seq Data Analysis

High-quality sequencing data in FASTQ format underwent initial processing using the fastp tool to remove adapter sequences, poly-N regions, and low-quality reads. After quality control, the remaining clean reads were assessed for their Q20 and Q30 scores and for their GC content. The reads were subsequently aligned to the reference genome using HISAT2 (v2.0.5), followed by novel transcript prediction with StringTie (v1.3.3b). Gene-level read quantification was performed using featureCounts, and gene expression levels were normalized to fragments per kilobase of transcript per million mapped reads (FPKM), which adjusts for both gene length and sequencing depth.

For differential expression analysis, the DESeq2 package (v1.20.0) in R was used to compare two experimental groups, each containing two biological replicates. This method employs a negative binomial model to identify statistically significant changes in gene expression. To control for false positives, the Benjamini–Hochberg procedure was applied to adjust the *p* values. Differentially expressed genes (DEGs) were defined as those with an adjusted *p* ≤ 0.05.

The functional annotation of the DEGs was performed using Gene Ontology (GO) enrichment analysis via the clusterProfiler R package 4.5.1. Additionally, KEGG pathway analysis was conducted to identify relevant biological pathways.

## 3. Results

### 3.1. Elevation Gradient Shaping the Gene Expression Patterns of Dendrobium nobile

To investigate how micro-elevation gradients influence gene expression in *Dendrobium nobile*, four semi-wild cultivation bases located at varying slope elevations in the Chishui region were selected: Yunji base (YJ01Y; 105°89′ E, 28°61′ N, 730 m), Chiyan base (CY01Y; 105°58′53″ E, 28°44′24″ N, 670 m), Kaixuan base (KX01Y; 105°76′ E, 28°45′ N, 548 m), and Zhuanshi base (ZS01Y; 105°76′ E, 28°45′ N, 347 m). One-year-old *D. nobile* stems were collected from each site, immediately frozen in liquid nitrogen to preserve RNA integrity, and subjected to RNA sequencing.

Using the *Dendrobium nobile* Lindl. genome (NCBI: GCA_022539455.1_Dnobile, https://ftp.ncbi.nlm.nih.gov/genomes/all/GCA/022/539/455/GCA_022539455.1_Dnobile/) (accessed on 20 June 2023) as a reference, a total of 31,331 gene sequences were identified, including 1855 novel genes. Principal component analysis (PCA) (Figure 1A) revealed that although there was no absolute clustering separation across samples, especially with significant overlap between the YJ01Y and CY01Y samples, there was notable spatial separation between the low-elevation base (ZS01Y) and the high-elevation bases (YJ01Y and CY01Y) along Principal Component 1 (PC1), which explained 16.73% of the variation. Notably, the KX01Y (middle elevation) samples fell between CY01Y and ZS01Y on the PC1 axis, and the gradient distribution of this sample was consistent with the actual elevation, reflecting the continuous regulatory effect of elevation on the gene expression patterns of *D. nobile*. Additionally, the PCA results also indicated that there was a certain degree of natural genetic variation among *D. nobile* individuals, which might explain some of the transcriptional differences observed.

Further comparative analysis of the gene expression profiles confirmed the impact of elevation (Figure 1B). A total of 14,647 shared expressed genes were identified across all four semi-wild cultivation bases. The high-elevation bases (YJ01Y and CY01Y) each had 264 uniquely expressed genes, whereas the middle-elevation base KX01Y and the low-elevation base ZS01Y had 232 and 488 uniquely expressed genes, respectively. This base-specific expression profile clearly indicated that different elevation environments induced significant transcriptional differences in the host plant.

The screening of differentially expressed genes (DEGs) (Figure 1C) quantified this variation. Among these DEGs, the greatest number of DEGs was found between CY01Y (high elevation) and ZS01Y (low elevation), with 4833 DEGs, followed by YJ01Y (high elevation) and ZS01Y (4073 DEGs). In contrast, the difference between KX01Y (middle elevation) and ZS01Y was the smallest, with 1209 DEGs.

Given the primary focus of this study on the impact of elevation on the content of dendrobine, a key medicinal component in *D. nobile*, we also measured the dendrobine content in the samples from each base (Figure 1D). The results revealed that CY01Y (670 m) had the highest dendrobine accumulation, whereas ZS01Y (347 m) had the lowest, which is consistent with previous research [4]. On the basis of the observed elevation-dependent variations in both the DEGs and the dendrobine content, identifying gene sequences that exhibit covariation with both elevation and dendrobine synthesis will be the focus of further in-depth research.

### 3.2. Plant Hormone Signal Transduction as the Core Regulatory Pathway in the Elevation Response

To systematically analyze the functional significance of the gene expression differences across samples from different elevation gradients, we performed KEGG pathway enrichment analysis on DEGs from key base pair combinations (Figure 2). In the comparison between high-elevation CY01Y and YJ01Y, a total of 599 DEGs were identified. Among them, 36 genes were significantly enriched in the MAPK signaling pathway (ath04016), and 39 genes were enriched in the plant hormone signal transduction pathway (ath04075). Among the hormone signaling genes, 25 were upregulated, and 14 were downregulated (Figure 2A). These findings suggest that even at similar high elevation levels, significant differential responses in hormone signaling pathways exist.

In the comparison between the highest elevation (CY01Y) and the lowest elevation (ZS01Y), a total of 1087 DEGs were obtained. Among them, 36 genes were enriched in the glutathione metabolism pathway (ath00480), and 64 genes were significantly enriched in the plant hormone signal transduction pathway (ath04075). In this pathway, 44 genes were upregulated, and 20 were downregulated (Figure 2B). These results indicate that plant hormone signaling pathways undergo strong differential regulation under extreme elevation differences and constitute one of the core pathways driving adaptive changes in *D. nobile*.

In the comparison between CY01Y and KX01Y, a total of 767 DEGs were detected. Among these, 58 genes were significantly enriched in protein processing in the endoplasmic reticulum pathway (ath04141), and 38 genes were significantly enriched in the plant hormone signal transduction pathway (ath04075), with 29 genes upregulated and 9 genes downregulated (Appendix A). This result further highlights the role of hormone signal transduction in regulating transitions between different elevation zones.

In other base combinations (KX01Y vs. ZS01Y, YJ01Y vs. KX01Y, YJ01Y vs. ZS01Y), DEGs were enriched primarily in plant–pathogen interaction pathways (ath04626) and the upstream pathway of jasmonic acid (JA) biosynthesis—alpha–linolenic acid metabolism (ath00592). These findings suggest that, under nonextreme elevation differences, plant–pathogen interactions and lipid-derived signaling pathways may play a more prominent role in elevation adaptation.

Taken together, these results point to the plant hormone signal transduction pathway (ath04075) as a key regulatory axis underlying elevation-induced transcriptional responses in *D. nobile*. This pathway was consistently and significantly enriched across different elevation comparisons, with evident changes in gene expression patterns. It integrates multiple hormone signaling networks that regulate plant growth and metabolism, including auxin (Aux), cytokinin (CK), gibberellin (GA), abscisic acid (ABA), ethylene (ETH), brassinosteroid (BR), jasmonic acid (JA), and salicylic acid (SA) [14]. By integrating the expression profiles of key hormone pathway genes across each comparison group, we found that elevation changes, particularly the comparison between CY01Y and other bases, significantly regulated the expression of GA, ABA, JA, and SA signaling pathways, whereas the regulatory effects on the Aux, CK, ETH, and BR pathways were more scattered or not significant. Notably, the CY01Y samples at the highest elevation presented a significant and consistent increase in the regulation of these key hormone pathways compared with the other samples (Figure 3).

Gibberellins, a class of diterpenoid plant hormones, play critical roles in regulating plant growth and development, including seed germination, stem elongation, and stress responses [15]. GA recognition relies on the soluble receptor protein GID1, which binds bioactive GA and promotes the ubiquitination and degradation of DELLA proteins, thereby relieving growth repression. DELLA proteins, which belong to the GRAS family of transcription factors, act as negative regulators in the GA signaling pathway. Their stabilization inhibits plant growth but concurrently enhances stress tolerance and promotes the synthesis of secondary metabolites [16]. In this study, while the expression of the GID1 receptor gene did not significantly differ among samples from different elevations, the *DELLA* gene was significantly upregulated in the high-elevation CY01Y samples (Figure 4A). This expression pattern suggests that DELLA may function outside the traditional GA growth-promoting pathway, potentially participating in stress responses and secondary metabolite accumulation. Given that other canonical downstream GA growth-regulatory genes were not markedly upregulated, we hypothesize that DELLA may engage in crosstalk with other hormone pathways, such as jasmonic acid (JA), possibly through competitive interactions with JAZ proteins, thereby modulating JA signal transduction [17].

Abscisic acid is a major stress-responsive hormone that rapidly accumulates under drought, cold, and salinity stresses, regulating the expression of defense-related genes [18]. Our findings revealed that the ABA receptor gene *PYR*/*PYL* was significantly upregulated in CY01Y samples (Figure 4B), along with its downstream positive regulator, *SnRK2* (Figure 4C). This signal activation pattern implies that plants at CY01Y may experience moderate environmental stress, trigger the ABA pathway and subsequently enhance stress-responsive metabolism and the biosynthesis of functional metabolites such as dendrobine.

JA, a lipid-derived phytohormone, is known to regulate various physiological processes, including seed germination, root growth, flowering, senescence, and fruit ripening. It also plays a vital role in both biotic and abiotic stress responses, such as wounding, herbivory, and pathogen attack [19,20]. Our previous research revealed that lipid metabolism in plants tends to be more active at relatively high slope elevations [4]. In the present study, key components of the JA signaling pathway—*COI1*—were significantly upregulated (*p* < 0.05) in the CY01Y samples, *JAZ*, and *MYC2* showed an upward trend but the difference was not significant (*p* > 0.05) (Figure 4D–F), indicating that increased JA signaling contributes to the transcriptional activation of defense-related genes and secondary metabolite accumulation.

Salicylic acid is a small phenolic signaling molecule found widely in both plants and microbes. It participates in multiple aspects of plant development and plays essential roles in the response to environmental stress [21]. *NPR1* is a central regulatory component of the SA signaling pathway and the expression was not significantly upregulated (*p* > 0.05), but it has a promoting effect for the whole pathway (Figure 4G). Mutations in NPR1 are known to abolish SA-induced defense gene expression and disease resistance [22]. NPR proteins and catalase 2 (CAT2) do not directly bind DNA but regulate downstream gene expression via transcription factors. Among them, the TGA transcription factor was also highly expressed in the CY01Y samples, further supporting the activation of the SA pathway.

Together, these findings demonstrate that plant hormone signaling pathways clearly differ in terms of transcriptional regulation across the elevation gradient, with the GA, ABA, JA, and SA pathways showing the most consistent and significant upregulation at the high-elevation CY01Y base. These pathways are critically involved in plant defense responses, environmental adaptation, and secondary metabolism, strongly supporting a potential causal link between the “elevation gradient → hormone signaling → dendrobine accumulation”.

However, hormone signaling is influenced not only by external environmental cues but also by internal microenvironmental regulation. A growing body of evidence suggests that symbiotic relationships between plants and their endophytic microbiota can modulate hormone biosynthesis and signal transduction either directly or indirectly [23]. The plant microbiome has been increasingly recognized as a key regulatory node in hormonal responses. In model plants such as maize and *Arabidopsis thaliana*, endophytic fungi can influence the GA, JA, and ABA pathways and participate in the resistance architecture of plants. On the basis of these findings, we further investigated the structure of endophytic fungal communities in *D. nobile* at different elevations under semiwild cultivation conditions. Our goal was to elucidate the interactive relationship between microbial communities and plant hormone signaling pathways and to elucidate the potential mechanisms by which this interaction regulates dendrobine biosynthesis.

### 3.3. Structural Stability and Functional Differentiation of Endophytic Fungal Communities in Dendrobium nobile Along an Elevation Gradient

To investigate the impact of elevation on the endophytic fungal community within the stem tissues of *Dendrobium nobile*, we performed high-throughput sequencing targeting the ITS1F region across 30 samples from four semi-wild cultivation bases. After rigorous quality control, denoising, and removal of redundant sequences, a total of 1351 amplicon sequence variants (ASVs) were identified. Taxonomic annotation revealed 148 cores shared ASVs across all four cultivation sites (Figure 5A). The number of site-specific ASVs was 116 for YJ01Y (730 m), 169 for CY01Y (670 m), 157 for KX01Y (548 m), and 401 for ZS01Y (347 m). This distribution suggests that while elevation indeed shapes a portion of unique fungal compositions, a substantial proportion of the core microbiota is conserved across sites, forming a relatively stable microbial germplasm reservoir.

Alpha diversity analysis indicated no significant differences in fungal species richness or evenness across elevations (*p* > 0.05) (Figure 5B), suggesting overall microbial community stability across the semi-wild cultivation environments in Chishui.

To explore the phylogenetic structure of the endophytic fungal communities, we constructed a phylogenetic tree using representative sequences from the top 100 genera in terms of relative abundance (Figure 5C). The results of the abundance analysis revealed that the fungal communities were predominantly composed of Ascomycota, followed by Basidiomycota, with minor contributions from other phyla, including Olpidiomycota, Mortierellomycota, Mucoromycota, and Chytridiomycota. Notably, no significant phylogenetic clustering or specialization was observed across elevations, indicating that the evolutionary relationships of endophytic fungi in this region are highly conserved. This reflects long-term ecological selection pressures contributing to the stability of microbial community composition.

Beta diversity analysis based on the Bray–Curtis distance further supported these observations (Figure 5D). The overall community similarity among sites did not exhibit a clear elevation-dependent turnover pattern. Nonmetric multidimensional scaling (NMDS) (stress < 0.1) revealed substantial overlap among samples from the four bases (Figure 5E), with only slight deviations observed in three samples each from KX01Y and ZS01Y. Collectively, these results suggest that the overall structure of endophytic fungal communities in *D. nobile* remains highly stable across different elevations, with the lowest community heterogeneity observed in the high-elevation CY01Y and YJ01Y samples, indicating a potentially more robust plant–microbe interaction network at these sites.

Although the above findings highlight structural stability, they do not necessarily imply functional homogeneity. To investigate this further, we employed similarity percentage (SIMPER) analysis to identify key functional taxa that drive subtle community differences under a background of overall stability. Fungal genera with a contribution ratio > 5% were retained.

The results showed that the differences between CY01Y (high) and KX01Y (midelevation) were driven primarily by *Olpidium*, *Saccharomycopsis*, *Pseudopithomyces*, *Plectosphaerella*, and *Rhizopus* (Figure 6A). The major differentiating taxa between CY01Y (high) and ZS01Y (low) were *Olpidium*, *Plectosphaerella*, *Pseudopithomyces*, *Strelitziana*, and *Cladosporium* (Figure 6B). For the CY01Y (high) vs. YJ01Y (high) comparison, the key contributors were *Hannaella*, *Olpidium*, *Strelitziana*, *Carlosrosaea*, and *Pseudopithomyces* (Figure 6C). In summary, although the overall community structure remained stable, the relative abundances of specific functional fungal genera varied significantly with elevation. Among these, *Olpidium*, *Plectosphaerella*, *Pseudopithomyces*, and *Strelitziana* were identified as core functional taxa driving elevation-responsive differences.

In-depth analysis revealed that *Olpidium* spp. were consistently present across all the sites (Appendix A), with representative species including *O. brassicae* and *O. virulentus*. *Olpidium* fungi are considered among the earliest terrestrial fungi with motile zoospores and typically act as obligate intracellular parasites [24,25]. *O. brassicae* is known to infect root epidermal cells of cruciferous plants [26]. This study is the first to report its colonization of the stem tissues of orchidaceous species such as *D. nobile*.

In terms of elevation-specific functional taxa, the high-elevation CY01Y samples were enriched in *Pseudopithomyces* spp. (e.g., *P. chartarum*, *P. rosae*), *Setophoma* spp. (*S. yingyisheniae*, *S. chromolaenae*), *Phaeosphaeria* spp. (*P. sinensis*, *P. lunariae*, *P. oryzae*), and *Selenophoma* spp. (*S. mahoniae*). In contrast, YJ01Y was dominated by *Hannaella* spp. (e.g., *H. pagnoccae*, *H. sinensis*, *H. surugaensis*, *H. oryzae*, and *H. luteola*). This distribution pattern suggests that *D. nobile* may adopt selective microbial recruitment strategies at different elevations to modulate environmental sensing and metabolic responses.

### 3.4. Elevation-Dependent Functional Variations in Endophytic Fungal Communities Reveal Microbial-Mediated Ecological Effects

To reveal the ecological functions of the endophytic fungal communities in the stems of *Dendrobium nobile* and their responses to elevation gradients, we performed functional prediction analysis on the FUNGuild database for the annotated fungal taxa [27]. FUNGuild classifies fungal communities into three main nutritional modes: pathotrophs, symbiotrophs, and saprotrophs.

The functional prediction of the endophytic fungal communities revealed that while all four cultivation bases shared core functional groups, including animal and plant pathogens, endophytes, and ectomycorrhizal fungi (Figure 7A), the functional weights varied significantly across the elevation gradient. The lowest elevation (ZS01Y, 347 m) presented the broadest functional group distribution (Figure 7B), which was dominated by plant pathogens, animal pathogens, and ectomycorrhizal fungi. At middle elevations (KX01Y, 548 m), the functional groups were more concentrated, with undefined saprotrophs and plant pathogens as the dominant types. In contrast, the high-elevation sites (CY01Y, 670 m, and YJ01Y, 730 m) displayed a distinct saprotrophic–parasitic dual-drive pattern. Specifically, CY01Y was dominated by undefined saprotrophs, fungal parasites, and animal/plant pathogens (Figure 7C), whereas YJ01Y presented a greater abundance of plant pathogens, undefined saprotrophs, and fungal parasites. This functional variation clearly has ecological significance: saprotrophs promote nutrient mineralization through organic matter decomposition, whereas pathogenic and parasitic fungi activate plant immune systems (e.g., the JA/SA signaling pathways) to induce defensive metabolic responses.

Modern ecological theories confirm that plant–environment interactions are essentially mediated by the microbiome, encompassing processes such as nutrient cycling, stress adaptation, and secondary metabolism regulation [28]. The synergistic enrichment of saprotrophs and pathogens observed at the high-elevation sites provides critical evidence for explaining the formation of the geo-authentic quality with higher concentrations of active ingredients and more stable plant morphology of *D. nobile* in the Chishui region, especially the high dendrobine content at CY01Y. This microbial community-driven mechanism reshapes the host’s hormonal balance and defense metabolism network, ultimately mediating the influence of the elevation gradient on the medicinal quality of the plant.

## 4. Discussion

The interactions between plants and complex microbial communities stem from millions of years of co-evolution, with environmental factors playing crucial roles throughout this process. The unique Danxia landform in the Chishui region creates micro-elevation gradients at the spatial scale, driving the formation of differentiated plant–microbe interaction systems across various cultivation bases. Within this system, plant hormones serve as central mediators [29,30], connecting microorganisms and the host through bidirectional regulatory mechanisms. These hormones manipulate the host’s hormonal network using hormone mimics and protein effectors, conversely, plant-derived hormones can directly or indirectly regulate microbial metabolic activities and community assembly processes through signal cascades [31]. The results of this study indicate that the micro-elevation differences between cultivation bases influence dendrobine biosynthesis through a dual-pathway approach by directly activating hormone pathways and reshaping the endophytic fungal community structure. These pathways synergistically interact within the hormone signaling network of *D. nobile*, ultimately regulating the biosynthesis of the key stress-resilient metabolite dendrobine.

Importantly, while each plant hormone triggers specific molecular pathways, its effectiveness is expanded through hormone crosstalk [30] (Figure 8), forming a dynamic regulatory network that optimizes the adaptability of the plant to environmental changes. Functional prediction analysis further revealed that the high-elevation site CY01Y was significantly enriched with plant pathogens and fungal parasites. These groups play dual ecological roles in the microecosystem, potentially harming individual plants in the short term. However, over the long term, they regulate population dynamics, promote biodiversity, drive co-evolution, and participate in material cycling, thus becoming key biotic factors that maintain the ecosystem’s function and stability. As explained by the Janzen–Connell hypothesis, pathogens and parasites exert “density-dependent inhibition” to limit the expansion of dominant species, creating ecological niches that promote community diversity. Moreover, the persistent selective pressure they impose drives plants to evolve disease resistance mechanisms, ultimately increasing genetic diversity and environmental adaptability in plant populations over time [32].

### 4.1. Correlation Between Hormone-Related Gene Expression and Environmental Changes

This study revealed that the expression of key genes involved in plant hormone signaling pathways in *Dendrobium nobile* plants from different cultivation bases clearly differed in an elevation-dependent manner. Notably, at the high-elevation site CY01Y (670 m), several core regulatory factors related to hormone signaling tended to be synergistically upregulated, including the gibberellin pathway repressor DELLA, abscisic acid receptor PYR/PYL, kinase SnRK2, core components of the jasmonic acid pathway (COI1-JAZ-MYC2), and the salicylic acid regulator NPR1-TGA, all of which were co-expressed in an upregulated manner. This pattern of coordinated expression suggests that high-elevation environments may induce plants to activate a stronger stress-defense signaling network, which is significantly positively correlated with dendrobine accumulation. Notably, complex cross-regulation (hormonal crosstalk) occurs between hormone pathways, meaning that not all hormone pathways are consistently activated or suppressed under all environmental conditions. To further validate the functional output of the gene expression differences, we measured the actual jasmonic acid (JA) levels in the samples. The results revealed that the JA levels in *D. nobile* from CY01Y were significantly higher than those from the low-elevation bases (Appendix A), supporting transcriptional upregulation at the physiological level and thereby strengthening the functional link between “gene → hormone → metabolism.” These findings highlight the ability of external environmental factors, particularly abiotic factors associated with elevation, to shape plant hormone networks [33]. The environmental factor combinations among the different cultivation bases in the Chishui region differ significantly across the micro-elevation gradient. By reprogramming the expression of hormone-related genes, *D. nobile* optimizes its growth and defense responses across various ecological niches. This regulation of hormone pathways shapes its unique adaptive phenotypic plasticity, which is a key strategy for plant survival and evolution in heterogeneous environments.

### 4.2. Endophytic Fungal Populations and Co-Regulation of Metabolite Synthesis by Plant Hormones

The composition and structure of the endophytic fungal communities in *D. nobile* plants from different cultivation bases also clearly changed. As a crucial component of internal plant homeostasis, the microbiome establishes complex symbiotic interactions with the host. Extensive evidence suggests that endophytic fungi can influence plant hormone metabolism and signal transduction through multiple pathways [34]. Some fungal strains produce secondary metabolites that resemble plant endogenous hormones, directly intervening in the regulation of hormone signaling pathways. Moreover, endophytic fungi can indirectly affect the plant’s hormone balance by regulating the supply of hormone biosynthesis precursors, the activity of key synthetic enzymes, hormone transport, and degradation processes.

In this study, the differences in microbial communities across the four cultivation bases were shown to intervene in the regulation of *D. nobile*’s hormone network through the aforementioned pathways, driving the biosynthesis of key metabolites such as dendrobine in coordination with gene expression changes (Figure 9). This sustained signaling dialog between microorganisms and plants forms a highly integrated multilevel regulatory system, allowing *D. nobile* to sense and integrate external environmental stimuli and internal physiological signals, achieving precise regulation of hormonal homeostasis.

This regulatory mechanism ultimately drives plants to form specific adaptive metabolic responses under different elevation-induced stresses, such as increased accumulation of dendrobine. The involvement of endophytic microorganisms significantly enhances the response speed and plasticity of the plant’s hormone regulation network, providing the host with a more flexible “ecological regulatory mechanism.” These findings also provide a biological foundation for long-term adaptive evolution in heterogeneous environments.

### 4.3. Complexity and Synergy of Hormone Signal Transduction Networks

The regulatory effects of plant hormones do not arise from the independent activation of a single pathway but rather from the dynamic interweaving and synergistic regulation of multiple signaling pathways, forming a hormone crosstalk network. In *D. nobile*, this network involves the core components of the jasmonic acid (JA) pathway, JAZ proteins, and the transcription factor MYC2, which act as key nodes in regulating defense metabolism. This hub not only coordinates the expression of JA-responsive genes but also crosstalks with auxin, ethylene, and other pathways, forming multilevel cascading regulatory modules [35]. When *D. nobile* is grown in different environments, hormone changes induced by various environmental factors interact through this complex network, integrating multiple signal inputs and ultimately determining the synthesis of stress-resilient metabolites such as dendrobine. This synergistic hormone regulatory mechanism not only ensures the normal growth and development of *D. nobile* under different environmental conditions but also provides efficient and plastic physiological regulatory capabilities when facing complex abiotic stresses induced by elevation gradients. This highly cooperative hormone network allows *D. nobile* to form a flexible and robust stress-resilient phenotype, enhancing its geo-authentic medicinal quality.

### 4.4. Systemic Regulation of Environment-Microbe-Hormone-Metabolism Interactions

In conclusion, the planting environment, microbiome, and plant hormone metabolism system jointly influence the hormone pathways within *D. nobile* cells, thereby affecting dendrobine accumulation. By altering the ecological environment of a plant, external factors first trigger primary changes in plant hormone metabolism, which are then transmitted and amplified through the hormone signaling network. These changes activate or suppress a series of downstream transcription factors and target genes, including those closely related to the dendrobine synthesis pathway. Moreover, microbial members, which are internal factors, also play essential roles in this process. Microorganisms can directly participate in plant hormone metabolism, altering the actual available amount of hormones within the plant. Moreover, microorganisms may also sense and respond to changes in plant hormones, subsequently adjusting their own metabolic activities and interactions with the plant, forming a bidirectional feedback regulatory loop. This synergistic mechanism between external and internal factors enables *D. nobile* to dynamically adjust its hormone levels and metabolite synthesis patterns under complex environmental conditions across different cultivation sites, optimizing its growth and survival state.

## 5. Conclusions

This study systematically reveals a dual regulatory mechanism underlying dendrobine accumulation in *Dendrobium nobile* cultivated under semi-wild conditions at different slope elevations in the Chishui region. The findings demonstrate that an elevation of 670 m (CY01Y base) represents the optimal cultivation height for dendrobine accumulation. At this elevation, several plant hormones signaling pathways, including the GA, ABA, JA, and SA pathways, were significantly activated and closely associated with key metabolic pathways. Moreover, the endophytic fungal community at this elevation exhibited higher structural stability and compositional concentration and was predominantly composed of undefined saprotrophs, fungal parasites, and plant pathogens. Functional profiling revealed that these microbial functional guilds were significantly positively correlated with both hormones signaling and the dendrobine content. It is speculated that these fungi participate in plant–microbe interactions that enhance plant defense responses and secondary metabolism, thereby promoting dendrobine biosynthesis and accumulation. In summary, elevation-related microenvironmental differences not only reshaped the functional structure of the endophytic fungal community but also influenced the expression of hormonal signals in the host plant. This coordination occurs through a hormone–microbe interaction network, which ultimately drives the biosynthesis of defense-related secondary metabolites in *D. nobile*. These findings provide a theoretical basis for the precise cultivation of geo-authentic medicinal plants under semi-wild conditions and offer a novel perspective for understanding the regulatory mechanisms underlying secondary metabolite biosynthesis in medicinal plants.

## Figures and Tables

**Figure 1 biomolecules-15-01366-f001:**
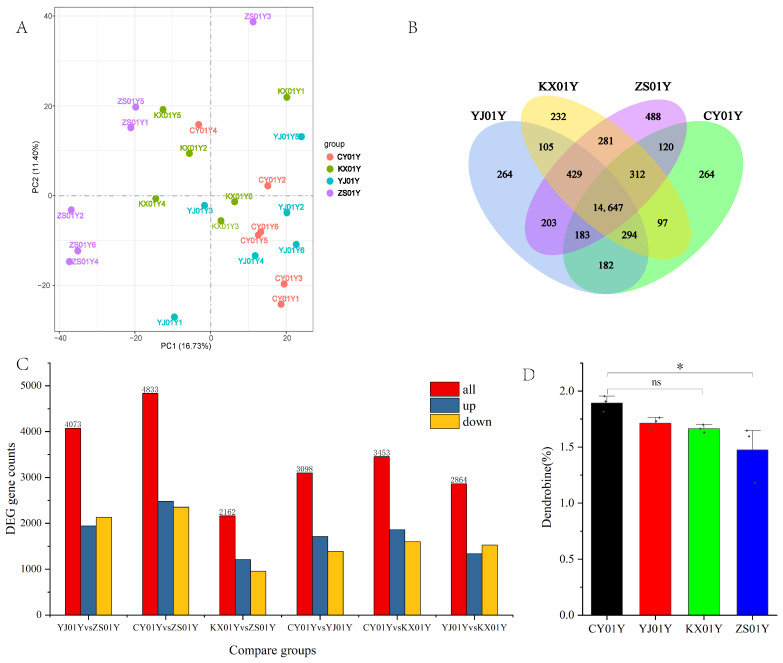
Transcriptomic profiling and dendrobine content variations in *Dendrobium nobile* stems cultivated at different hillside elevations. (**A**) Principal component analysis (PCA) of RNA-Seq data from four cultivation bases. The samples from CY01Y and YJ01Y (high elevation) clustered closely, whereas those from KX01Y (mid elevation) and ZS01Y (low elevation) were more dispersed, indicating transcriptional differences along the elevation gradient. (**B**) Venn diagram showing the number of shared and uniquely expressed genes across the four bases. A distinct base-specific transcriptional signature is observed. (**C**) Pairwise differential gene expression analysis. The greatest number of DEGs was observed between CY01Y and ZS01Y, whereas the fewest occurred between KX01Y and ZS01Y. (**D**) Dendrobium content (%) in the stem tissues of *D. nobile* from each base. The highest content was detected in CY01Y, and the lowest was detected in ZS01Y (* *p* < 0.05).

**Figure 2 biomolecules-15-01366-f002:**
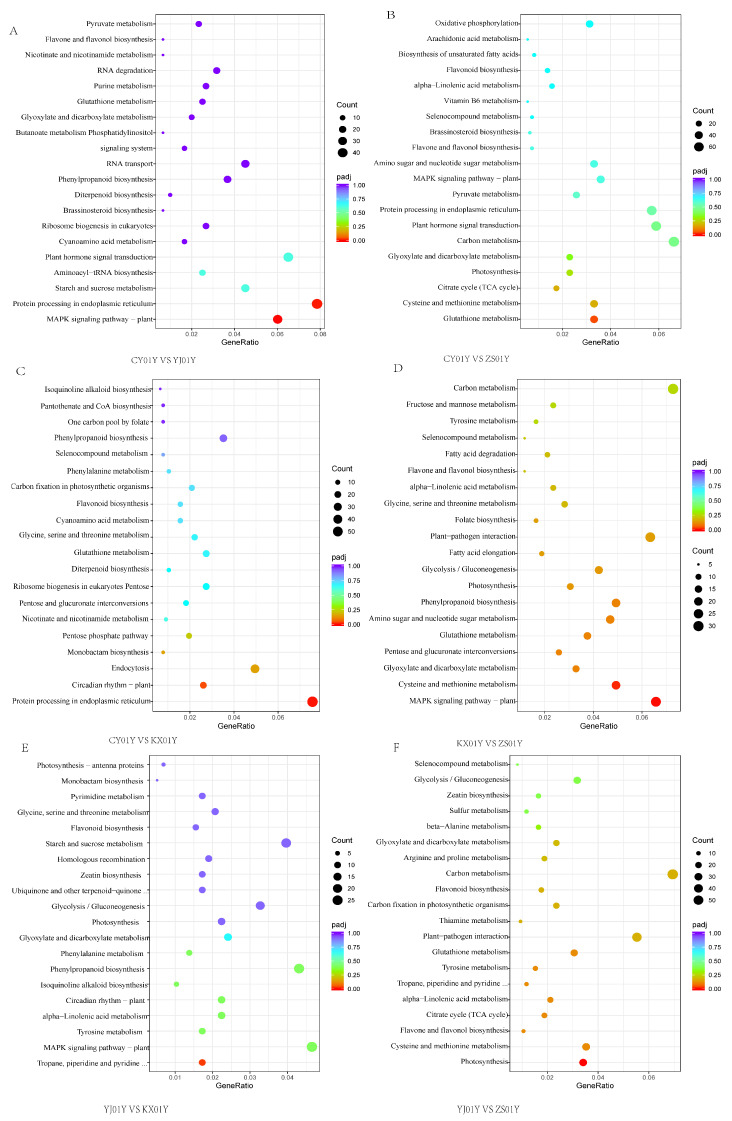
KEGG pathway enrichment analysis of DEGs in *D. nobile* across six pairwise base comparisons. (**A**) CY01Y vs. YJ01Y. (**B**) CY01Y vs. ZS01Y. (**C**) CY01Y vs. KX01Y. (**D**) KX01Y vs. ZS01Y. (**E**) YJ01Y vs. KX01Y. (**F**) YJ01Y vs. ZS01Y.

**Figure 3 biomolecules-15-01366-f003:**
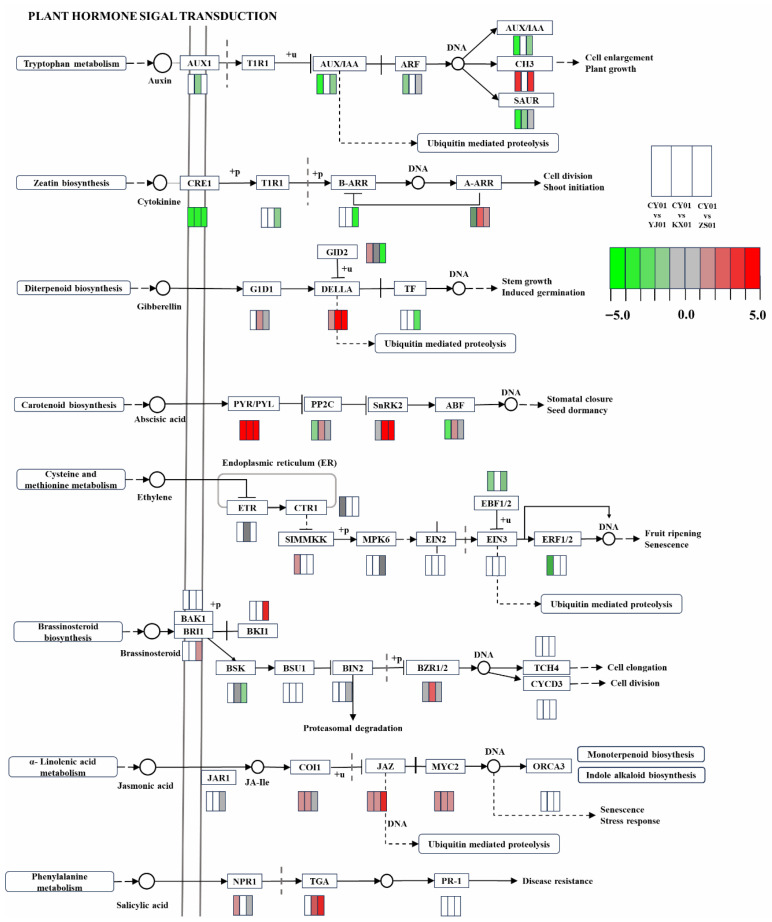
Differential gene expressions in plant hormone signal transduction pathways among *D. nobile* samples from four elevations. Heatmap showing changes in the expression of key signaling genes across three comparisons: CY01Y vs. YJ01Y, CY01Y vs. KX01Y, and CY01Y vs. ZS01Y. Hormone-related genes in the GA, ABA, JA, and SA pathways exhibited coordinated upregulation in CY01Y.

**Figure 4 biomolecules-15-01366-f004:**
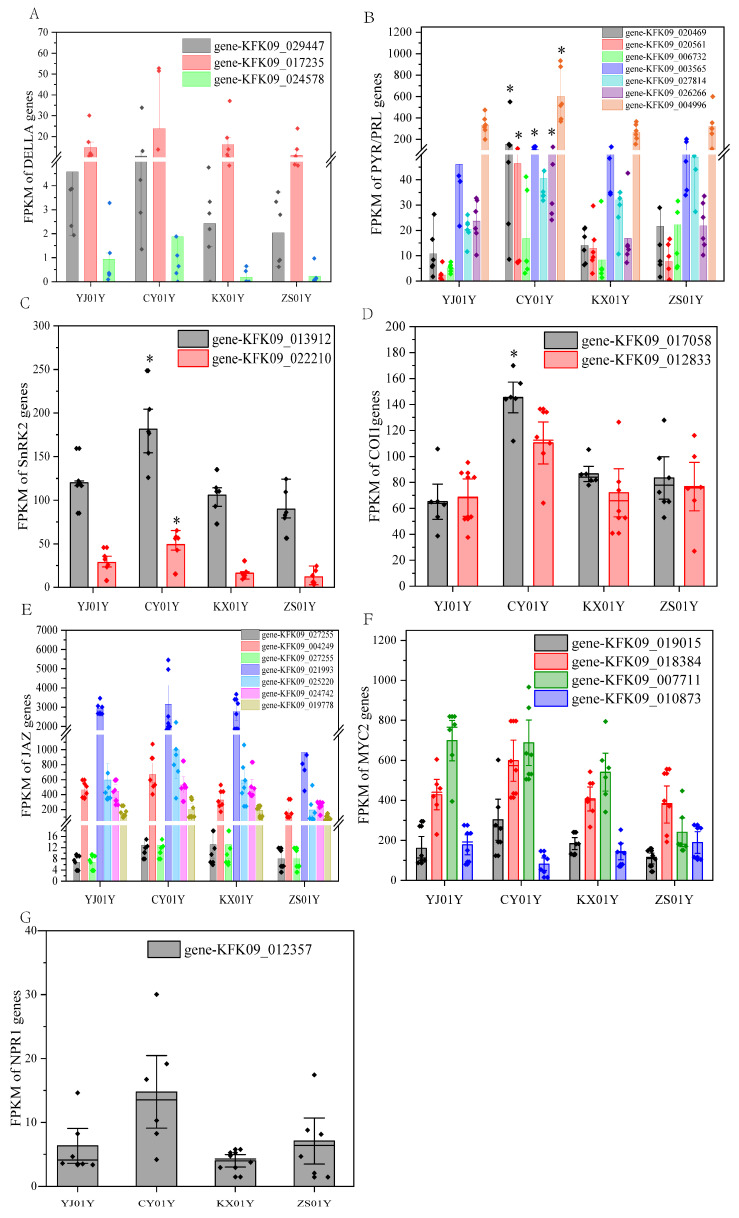
Expression profiles of representative hormone signaling genes in *D. nobile* from four cultivation sites. (**A**) DELLA (GA pathway) expression: three DELLA-encoding genes were significantly upregulated in CY01Y (*p* < 0.05). (**B**) PYR/PYL receptors (ABA pathway): all seven PYR/PYL genes presented significantly increased expression in CY01Y (*p* < 0.05). (**C**) SnRK2 (ABA pathway): Both SnRK2 genes were significantly upregulated in CY01Y (*p* < 0.05). (**D**) COI1 (JA pathway): One COI1 gene was significantly upregulated in CY01Y (*p* < 0.05). (**E**) JAZ (JA pathway): expression in CY01Y was elevated but not statistically significant (*p* > 0.05). (**F**) MYC2 (JA pathway): gene expression tended to be greater in CY01Y but is not significantly (*p* > 0.05). (**G**) NPR1 (SA pathway): elevated expression was observed in CY01Y but is not significantly (*p* > 0.05). The meaning of “*” is significantly higher than other groups. The diamonds represents data points.

**Figure 5 biomolecules-15-01366-f005:**
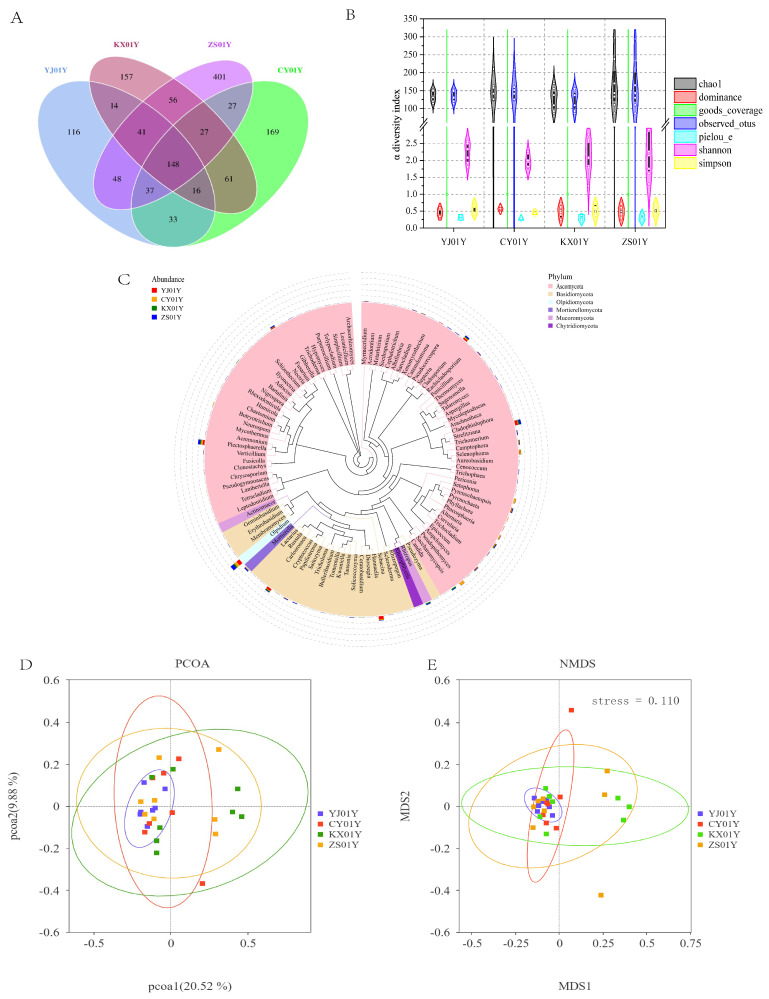
Diversity and phylogenetic structure of endophytic fungal communities in *D. nobile* stems from four cultivation bases. (**A**) Venn diagram showing shared and unique fungal ASVs across YJ01Y, CY01Y, KX01Y, and ZS01Y. (**B**) Alpha diversity metrics (richness and evenness) indicate no significant differences among sites. (**C**) Phylogenetic tree of the top 100 genera by relative abundance; outer ring bars represent genus-level abundance across samples. (**D**) Principal coordinate analysis (PCoA) based on the Bray–Curtis distance via unweighted UniFrac. (**E**) Nonmetric multidimensional scaling (NMDS) indicating high overlap among samples; only minor shifts were observed in KX01Y and ZS01Y.

**Figure 6 biomolecules-15-01366-f006:**
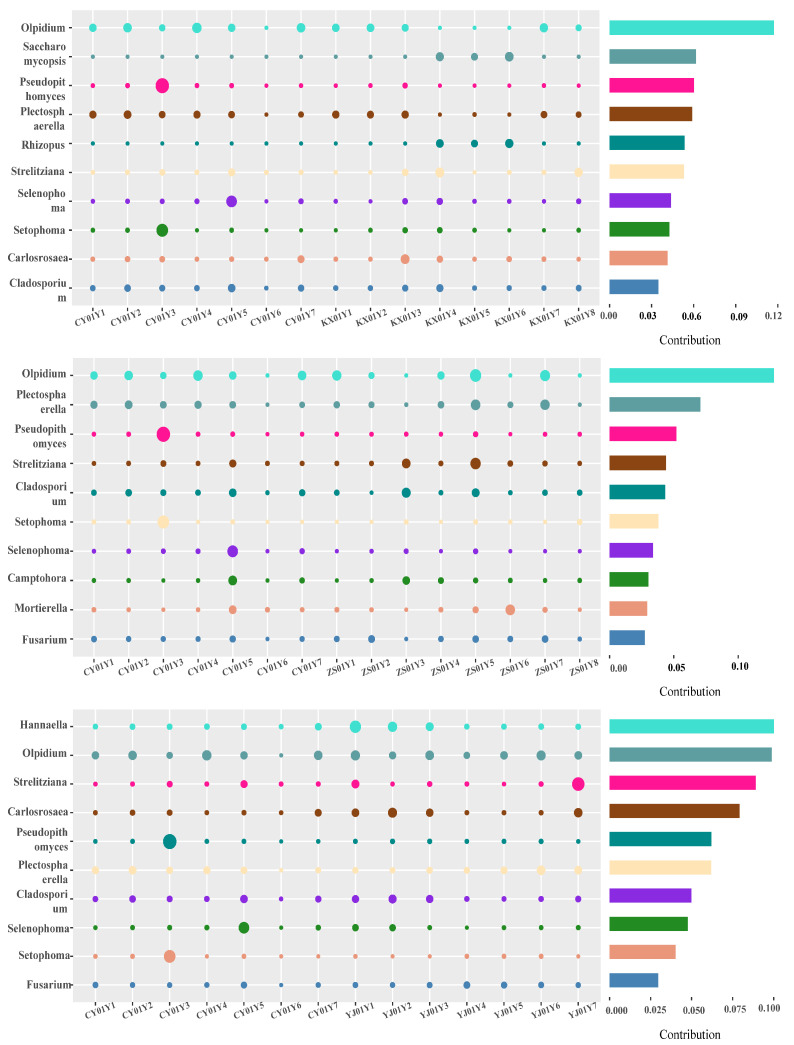
SIMPER analysis identifying key endophytic fungal genera driving compositional differences between CY01Y and other bases. The larger of the circle size, the higher the expression of the microorganism in this plant sample. (**A**) CY01Y vs. KX01Y. (**B**) CY01Y vs. ZS01Y. (**C**) CY01Y vs. YJ01Y.

**Figure 7 biomolecules-15-01366-f007:**
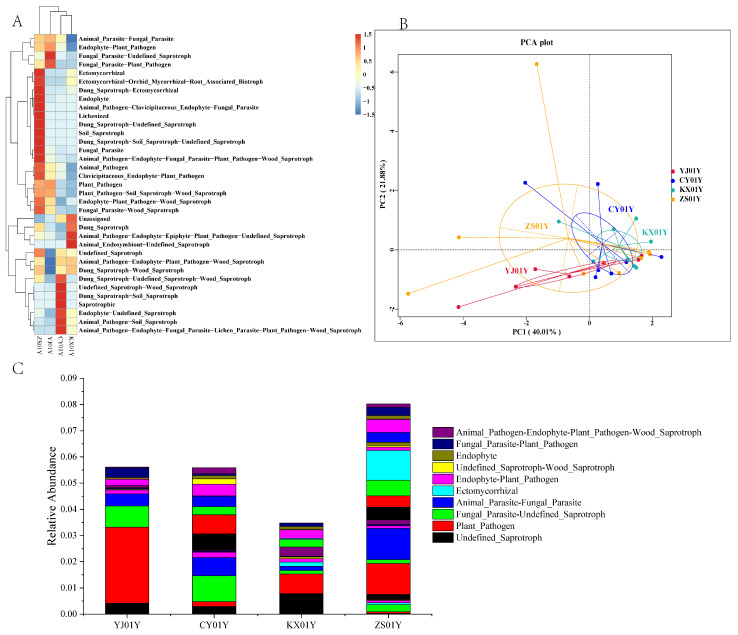
Functional guild prediction of endophytic fungal communities using FUNGuild. (**A**) Heatmap of the predicted trophic modes of the four fungal communities. (**B**) Principal component analysis (PCA) based on functional group composition. (**C**) Relative abundance bar plot of major predicted functional guilds. High-elevation sites show coenrichment of saprotrophs and plant pathogens.

**Figure 8 biomolecules-15-01366-f008:**
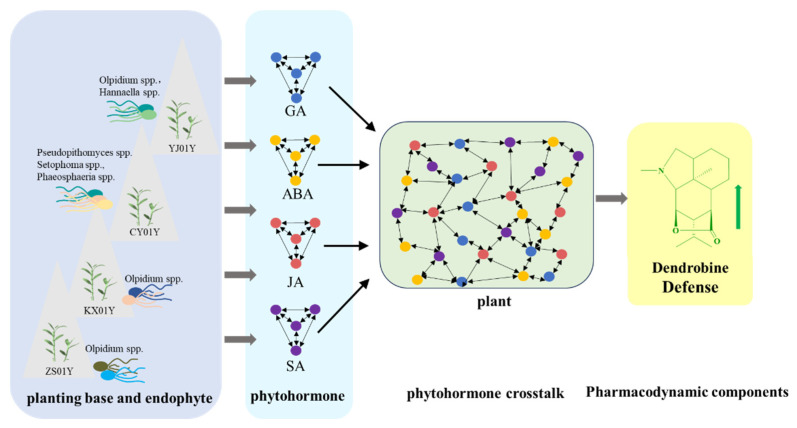
Schematic diagram illustrating hormone crosstalk in *D. nobile* under elevation-driven environmental stress.

**Figure 9 biomolecules-15-01366-f009:**
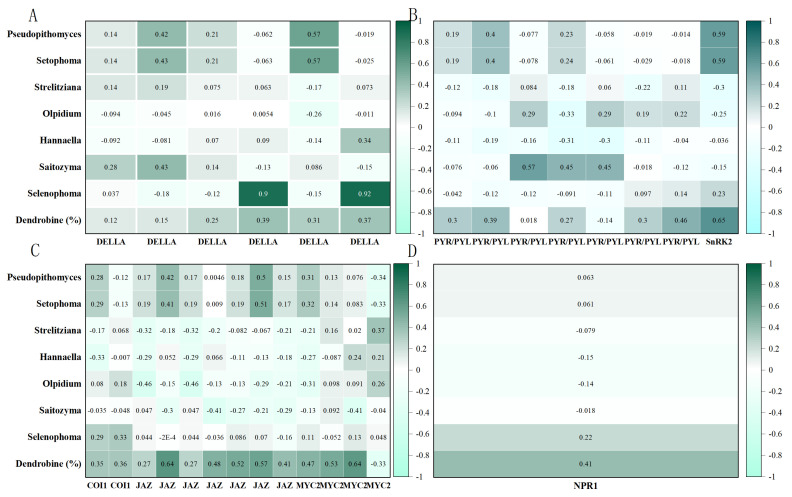
Correlations between key endophytic fungal genera and hormone-related genes involved in dendrobine biosynthesis in *D. nobile*. (**A**) GA pathway. (**B**) ABA pathway. (**C**) JA pathway. (**D**) SA pathway. Color gradients indicate correlation strength, and values represent Spearman coefficients.

## Data Availability

The authors confirm that the data supporting the findings and conclusions of this study are available in the article.

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
