# Peer review of "A Dual Regulatory Mechanism of Hormone Signaling and Fungal Community Structure Underpin Dendrobine Accumulation in *Dendrobium nobile"

_biomolecules, 2025, doi:10.3390/biom15101366_

Round 1
Reviewer 1 Report
Comments and Suggestions for Authors
The manuscript “A Dual Regulatory Mechanism of Hormone Signaling and Fungal Community Structure Underpin Dendrobine Accumulation in Dendrobium nobile” presents an interesting multi-omics study. The integration of transcriptomics, fungal community profiling, and hormone analysis suggests valuable insights into how elevation influences dendrobine accumulation. The study is timely, the methods are carefully applied, and the overall presentation is clear and logically structured. I believe this work makes a meaningful contribution to the field and will be of interest to readers of Biomolecules.
I recommend acceptance after minor revision. My suggestions are small clarifications that will help strengthen the readability and precision of the manuscript:
- Line 25: Consider simplifying “dual-path regulatory mechanism” to “dual regulatory mechanism” for smoother readability.
- Line 442: Please add a brief explanation of the term “geo-authentic quality,” since not all readers may be familiar with this concept in the context of traditional Chinese medicine.
- Figure 4 (Lines 329–338): In cases where gene expression changes are not statistically significant (e.g., JAZ, NPR1), please make this clear in the figure legend and text. Currently they are described as “elevated,” which could be misleading.
Author Response
- Line 25: Consider simplifying “dual-path regulatory mechanism” to “dual regulatory mechanism” for smoother readability.
Answer: Thank you for your valuable comments and suggestions. We have made changes on the original manuscript according to your suggestion.
- Line 442: Please add a brief explanation of the term “geo-authentic quality,” since not all readers may be familiar with this concept in the context of traditional Chinese medicine.
Answer: Thank you for your valuable comments and suggestions. The “geo-authentic quality” refers to medicinal herbs cultivated in this region exhibiting superior quality, higher concentrations of active ingredients, greater yields, and generally more effective clinical outcomes. So, we have made changes on the original manuscript: “The synergistic enrichment of saprotrophs and pathogens observed at the high-elevation sites provides critical evidence for explaining the formation of the geo-authentic quality with higher concentrations of active ingredients and more stable plant morphology of D. nobile in the Chishui region, especially the high dendrobine content at CY01Y.”
- Figure 4 (Lines 329–338): In cases where gene expression changes are not statistically significant (e.g., JAZ, NPR1), please make this clear in the figure legend and text. Currently they are described as “elevated,” which could be misleading.
Answer: Thank you for your valuable comments and suggestions. The genes depicted in Figure 4 are all associated with the plant hormone pathway. Although the expression levels of some genes (such as JAZ, MYC2, and NPR1) did not show significant differences across the four groups, they exhibited an overall upward trend in the CY01Y sample. Given that our analysis is based on semi-wild cultivated samples rather than strictly controlled laboratory specimens, these non-significant trends remain highly meaningful and offer valuable insights. In response to your suggestions, we have provided explicit annotations in both the text and figure legend.

Reviewer 2 Report
Comments and Suggestions for Authors
The manuscript presents very interesting and comprehensive results on the influence of geographical altitude and fungi on Dendrobium nobile. The study is ambitious and potentially valuable, but the following issues should be addressed to improve clarity and reproducibility:
- Abstract – Please clarify what “CY01Y” refers to.
- Terminology – Throughout the manuscript, terms such as “semi-wild cultivation bases,” “planting-bases,” “four bases,” and “high-elevation bases” are unclear. Consider defining these terms precisely when first mentioned.
- Introduction – The phrase “quality of D. nobile” needs to be specified. Does this refer to chemical composition, pharmacological activity, yield, or other traits?
- Materials and Methods
Section 2.1: It should be clearly stated whether the plant material was collected from wild populations or cultivated under controlled conditions.
Section 2.2: Please specify the method used for RNA extraction, including the kit or protocol reference.
Section 2.3: The origin of the fungi is unclear. Were they isolated from plant stems or obtained from another source?
Author Response
Reviewers 2#
- Abstract – Please clarify what “CY01Y” refers to.
Answer: Thank you for your valuable comments and suggestions. CY01Y refers to the Dendrobium nobile was from the ChiYan base at 670 m elevation, one of the four semi-wild cultivation bases. We have made changes on the original manuscript according to your suggestion.
- Terminology – Throughout the manuscript, terms such as “semi-wild cultivation bases,”“planting-bases,”“four bases,”and“high-elevation bases”are unclear. Consider defining these terms precisely when first mentioned.
Answer: Thank you for your valuable comments and suggestions. The terms “semi-wild cultivation bases,” “planting-bases,” and “four bases” all convey the same meaning. Therefore, throughout the text, I have uniformly replaced them with “semi-wild cultivation bases” and clarified their definition. Additionally, the concept of “high-elevation bases” and low-elevation bases were explained in 2.1.
- Introduction–The phrase“quality of D. nobile”needs to be specified. Does this refer to chemical composition, pharmacological activity, yield, or other traits?
Answer: Thank you for your valuable comments and suggestions. The quality of medicinal materials encompasses, but is not limited to, yield, appearance, active ingredient content, and clinical medicinal value. Our article primarily focuses on the plant growth state and the content of active medicinal components, which have been clearly elaborated in the text.
- Section 2.1: It should be clearly stated whether the plant material was collected from wild populations or cultivated under controlled conditions.
Answer: Thank you for your valuable comments and suggestions. The plant samples are collected from semi-wild cultivation bases, not strictly controlled laboratory plants, nor pure plants. This has been clearly noted in the text.
- Section 2.2: Please specify the method used for RNA extraction, including the kit or protocol reference.
Answer: Thank you for your valuable comments and suggestions. We used the RNAprep Pure Plant Total RNA Extraction Kit (DP432) for RNA extraction and have summarized the procedure for RNA extraction.
Section 2.3: The origin of the fungi is unclear. Were they isolated from plant stems or obtained from another source?
Answer: Thank you for your valuable comments and suggestions. The endophytic bacteria were isolated from the stems of Dendrobium nobile and have been clearly described in the text.
